# Adverse Maternal Outcomes in Pregnant Women Affected by Severe-Critical COVID-19 Illness: Correlation with Vaccination Status in the Time of Different Viral Strains’ Dominancy

**DOI:** 10.3390/vaccines10122061

**Published:** 2022-11-30

**Authors:** Antonella Vimercati, Rosalba De Nola, Stefano Battaglia, Rossella Di Mussi, Gerardo Cazzato, Leonardo Resta, Maria Chironna, Daniela Loconsole, Lorenzo Vinci, Giulia Chiarello, Massimo Marucci, Ettore Cicinelli

**Affiliations:** 1Unit of Obstetrics and Gynecology, Department of Biomedical and Human Oncologic Science, University of Bari, Piazza G. Cesare 11, 70124 Bari, Italy; antonella.vimercati@uniba.it (A.V.); l.vinci@studenti.uniba.it (L.V.); g.chiarello4@studenti.uniba.it (G.C.); ettore.cicinelli@uniba.it (E.C.); 2Interdisciplinary Department of Medicine, University of Bari School of Medicine, Piazza G. Cesare 11, 70124 Bari, Italy; battagliastefano87@gmail.com; 3Department of Emergency and Organ Transplantation, Anaesthesia and Resuscitation Division, University of Bari, Piazza G. Cesare 11, 70124 Bari, Italy; rosselladimussi@libero.it (R.D.M.); massimo.marucci@tin.it (M.M.); 4Department of Emergency and Organ Transplant, Pathology Division, University of Bari, Piazza G. Cesare 11, 70124 Bari, Italy; gerycazzato@hotmail.it (G.C.); leonardo.resta@uniba.it (L.R.); 5Department of Interdisciplinary Medicine (Laboratory of Molecular Epidemiology and Public Health), University of Bari, Piazza G. Cesare 11, 70124 Bari, Italy; maria.chironna@uniba.it (M.C.); daniela.loconsole@uniba.it (D.L.)

**Keywords:** maternal, COVID-19, SARS-CoV2, variant, strains, VOC

## Abstract

This is a monocentric and cross-sectional study conducted at the COVID-19 Division of the Obstetrical and Gynecological Unit and Intensive Care Units (ICUs) of Policlinico di Bari, in Bari, Italy, between September 2020 and April 2022. This study aimed to identify the prevalence of severe-critical COVID-19 illness requiring access to the Intensive Care Unit (ICU) among 287 pregnant patients, and possible correlations between the SARS-CoV-2 variants, the specific pandemic wave (dominated by wild, Alpha, Delta, and Omicron strains), and severe-critical adverse maternal outcomes. The prevalence of severe-critical COVID-19 illness was 2.8% (8/287), reaching 4.9% (8/163) excluding the 4th wave (Omicron dominant). The Delta variant determined the highest risk ratio and odds for access to the ICU due to severe-critical COVID-19-related symptoms compared to the other variants (wild, Alpha, Omicron). During the third wave (Delta), the ICU cases underwent a higher rate of hyperimmune plasma infusion (75%), antibiotic therapy (75%), and remdesivir (33%); all of the patients were intubated. During the Omicron wave, the patients were asymptomatic or with few symptoms: most of them (70%) were vaccinated with a median of two doses. The maternal outcome worsened in the case of Alpha and, especially, Delta variants for severe-critical COVID-19-related symptoms and ICU access.

## 1. Introduction

The pandemic coronavirus disease (COVID-19) caused by the Severe Acute Respiratory Syndrome Coronavirus 2 (SARS-CoV-2), originated in December 2019, in Wuhan, Hubei Province, China. It spread quickly and widely. It was declared a Public Health Emergency of International Concern (PHEIC) by the Pan American Health Organization/World Health Organization (PAHO/WHO) on the 30 of January [1], covering four waves of the dominant viral variant until the present time. As of 15 April 2022, 503 million cases of COVID-19 have been reported in the world with more than 6.2 million deaths [2]. Like other RNA viruses, SARS-CoV-2 is characterized by random replication mutations that can influence infectiousness, virulence, and immune escape through new viral variants that are named with Greek letters by the World Health Organization (WHO) [2]. Among them, there are some variants of concern (VOCs), characterized by increased transmissibility or virulence and partial or total “resistance” to vaccines and antiviral drugs [2]. In July 2021, the Delta variant was identified in India. It became the dominant strain in November 2021, before nearly disappearing worldwide because of the new dominant VOC, the Omicron variant (or B.1.1.529) [2]. 

An emerging scientific interest is now growing in the sequence of placental COVID-19-related modifications with deciduitis, maternal-fetal malperfusion, fetal ipossic evolution, fetal immune and pro-inflammatory activation, severe neurological outcomes, and even congenital defects [3,4,5,6,7,8,9]. 

The present monocentric and cross-sectional study aimed to (1) identify the prevalence of severe-critical COVID-19 illness requiring the step-up into an ICU among all the admitted pregnant patients who delivered at our COVID-19 hub Center from August 2020 to April 2022; (2) find possible correlations amongst the SARS-CoV-2 variants, the specific pandemic wave (wild, Alpha, Delta, Omicron dominant), and the severe-critical adverse maternal outcomes, as recently shown by current literature [10,11,12,13,14]; (3) describe the possible differences between vaccinated and non-vaccinated patients during the fourth wave, as suggested by emerging data [15,16,17,18,19].

## 2. Materials and Methods

### 2.1. Study Design

This is a monocentric and cross-sectional study conducted at the COVID-19 Division of the Obstetrical and Gynecological Unit and Intensive Care Units (ICUs) of *Policlinico di Bari,* in Bari, Italy, between September 2020 and April 2022. The recruitment process had the following inclusion criteria: over the age of 18 years, hospitalized pregnant women during II and III trimesters with nasopharyngeal swabs positive for COVID-19 infection at the RT-PCR analysis, after informed consent, who delivered at our Centre. We included both vaccinated and non-vaccinated women, after informed consent, for comparison intention. The recruitment process had the following exclusion criteria: under the age of 18 years, pregnant outpatients at any gestational age with nasopharyngeal swabs positive for COVID-19 infection at the RT-PCR analysis who delivered elsewhere, hospitalized pregnant women during I trimester with nasopharyngeal swabs positive for COVID-19 infection at the RT-PCR analysis who delivered at our Centre. We did not consider patients in their first trimester due to the low number of cases and the rate of missing data about the delivery that was attended elsewhere. The present study was conducted according to the principles and standards of the “Good Clinical Research Practice” after the Ethical Committee’s approval and the acquisition of informed consent from all the patients. After the patients’ hospital discharge, their medical records were sanitized for data collection purposes.

The COVID-19 severity classification was based on the National Institutes of Health guidelines for disease severity, which gave detailed consideration of severe and critical scenarios, as reported below. [2]

*Severe Illness:* patients with SpO_2_ < 94% on room air at sea level, a ratio of arterial partial pressure of oxygen to fraction of inspired oxygen (PaO_2_/FiO_2_) < 300 mm Hg, a respiratory rate > 30 breaths/min, or lung infiltrates > 50%.

*Critical Illness:* patients with respiratory failure, septic shock, and/or multiple organ dysfunction (MOF). 

All respiratory samples were tested at the Laboratory of Molecular Epidemiology and Public Health of the Hygiene Unit (A.O.U.C. Policlinico Bari, Italy), which is the coordinator of the Regional Laboratory Network for SARS-CoV-2 diagnosis in the Apulia region, according to the procedure previously described [20]. The period between 1 August 2020 and 31 December 2020 was characterized by the circulation of non-VOC SARS-CoV-2 (wild dominant pandemic wave). Starting from December 2020 (Alpha dominant pandemic wave), the Alpha VOC was identified using the SGTF (S -gene Target Failure) as a robust proxy of B.1.1.7-lineage SARS-CoV-2 [20]. Starting from May 2021 (Delta dominant pandemic wave), all the SGTF-negative samples were supposed to be Delta VOC. The first Omicron VOC in the Apulia region was identified in December 2021. The variant showed a rapid spread in the whole region, reaching an estimated prevalence of 100% in a few weeks. Therefore, starting from January 2022, all the samples were supposed to be Omicron VOC (Omicron dominant pandemic wave).

The objectives of the study are listed below:

First objective: we identified 8 pregnant women admitted to ICU for severe-critical health issues related to COVID-19 out of 287 patients, independently from the pandemic wave and the viral strain (wild, Alpha, Delta, Omicron dominant).

Second objective: we compared 4 groups of patients depending on the pandemic wave (1st wave dominated by the wild variant, N 80; 2nd wave dominated by the Alpha strain, N 63; 3rd wave dominated by the subsequent VOC, Delta, N 20; 4th wave dominated by the VOC Omicron, N 124). We estimated the risk ratio for ICU access based on the specific pandemic wave and viral strain. 

Third objective: we evaluated the prevalence of vaccinated patients and their possible different symptoms—hospitalisation, ICU access, and treatment—in comparison with non-vaccinated woman.

The available vaccinations for pregnant woman were mRNA-based, in detail Pfizer or Moderna based on their availability at the vaccination hub.

### 2.2. Statistical Analysis 

The variables were numerical and factorial. The Shapiro-Wilk test and graphical evaluations of each numerical variable were performed to evaluate their correspondence with the normal distribution. The Bartlett test was performed to evaluate the homogeneity of variances within quantitative variables, but none of them were homoscedastic. A modified robust Brown-Forsythe Levene-type test, based on the absolute deviations from the median, confirmed the Bartlett test’s results. The categorical variables were illustrated as frequencies (%), whereas the numerical data were reported as a median and interquartile range [IQR] due to the absence of normal distribution and homoscedasticity. Statistical analyses were performed using the R statistical environment (The R Foundation for Statistical Computing; Vienna, Austria), specifically the packages “fBasics”, “graphics”, “ggplot2”, and “psych”. The relationship between categorical factors was evaluated with Fisher’s exact test (package “gmodels”). The estimated risk ratios were performed with the packages “epitools” and “epiR”. We calculated the risk ratio for ICU access by unconditional maximum likelihood estimation (Wald) and small sample adjustment (small) out of 287 pregnant patients, depending on their SARS-CoV2 variant or vaccination status. Confidence intervals were obtained by using normal approximation (Wald), normal approximation with small sample adjustment (small), and the bootstrap method (boot). Similarly, each row of the rx2 table was compared to the ICU access’ reference level and test of independence two-sided *p* values were calculated using Fisher’s Exact, Monte Carlo simulation, and the chi-square test. Plots and graphs were realized using the R package “graph”.

## 3. Results

### 3.1. Prevalence of Severe-Critical Maternal Outcomes and Possible Correlation with the Pandemic Waves and Viral Strains

The overall prevalence of severe-critical COVID-19 illness requiring the step-up into an ICU among the 287 pregnant patients (see Table 1) who delivered at our COVID-19 hub Center from August 2020 to April 2022 was 2.8%. This rate reached 4.9% during the pandemic waves dominated by wild, Alpha, and Delta variants, excluding the last one (Omicron dominant). During the fourth epidemic wave (Omicron prevalence), 70% of the patients were completely vaccinated and nobody accessed the ICU. 

The main characteristics of the eight severe-critical cases identified have been summarized in Table 2. 

Considering the eight severe-critical cases, the median maternal age was advanced (AMA) since it was >35 years old [21], and specifically 37.5 [IQR 4.5] years. Moreover, the median BMI was 30 [IQR 4.6], defining a condition of mild obesity [22]. Notably, two out of eight patients carried a twin pregnancy, and nobody had comorbidities (i.e., thyroid diseases, autoimmune diseases, hereditary thrombosis, diabetes, other chronic diseases), apart from mild obesity. All the patients were Caucasian. Half of the severe-critical cases of COVID-19 infection were sustained by the Delta variant of SARS-CoV2 during the third epidemic wave. Half of the patients were in their first pregnancy, but all of them delivered with a Cesarean Section because of COVID-19 severity, twins, and/or prematurity (see Table 3). 

All the patients described in Table 2 were admitted to the step-up unit (ICU) for severe-critical symptoms of COVID-19 infection. In detail, the overall Bilevel Positive Airway Pressure (BiPAP) ventilation rate was 75%, with a global oro-tracheal intubation (OTI) rate of 57%. The patients’ step-up stays had a median duration of 13.5 days [IQR 7.75], during which they were administered heparin (88%), both prophylactic and therapeutic; antibiotics (62%), especially cephalosporins and macrolides; hyperimmune plasma (62%); and only in one case of the Delta variant also the antiviral drug remdesivir (see Table 2 and Table 3).

From an obstetrical point of view, we observed a low gestational age (g.a.) with a median of 34.9 [IQR 3.72] and an average median neonatal weight percentile. The neonates accessed the step-down unit in 27% of cases and the step-up unit in 17% of cases. We reported a case of critical fetal-neonatal outcome that ended with an abortive cesarean section because of fetal intraventricular hemorrhage during the 23rd week of g.a.

Next, we focused on the clinical outcomes of the three variants (wild, Alpha, and Delta) responsible for the eight cases of severe-critical COVID-19 among pregnant patients as summarized in Table 3, Figure 1 and Figure 2.

The median maternal age and the median g.a. were higher in the wild group, respectively 41.5 [2.5] yrs and 37.1 [1.5] wks, ds. The Alpha group was characterized by the lowest median g.a., in detail 28.9 [5.35] wks, ds, and consequently also the lowest neonatal weight. Notably, all the cases affected by the Delta variant underwent OTI because of the clinical conditions due to COVID-19. Even the therapies during the step-up stays were different among the pandemic wave since the patients belonging to Delta dominant wave experienced a higher rate of hyperimmune plasma infusion (75%), antibiotic therapy (75%), and remdesivir (33%). 

### 3.2. Comparison between the SARS-CoV-2 Variants, the Specific Pandemic Wave (Wild, Alpha, Delta, Omicron Dominant), and the Severe-Critical Adverse Maternal Outcomes

Next, we compared four groups of patients (N 287) depending on the pandemic wave (1st wave dominated by the wild variant, N 80; 2nd wave dominated by the Alpha strain, N 63; 3rd wave dominated by the Delta variant, N 20; 4th wave dominated by the Omicron variant, N 124) and we estimated the risk ratio for ICU access based on the specific pandemic wave and the viral strain.

Notably, the Delta variant was correlated with a significantly higher ICU risk ratio (6.30) and ICU odds (0.25) as opposed to the wild variant (*p* 0.01), Alpha variant (*p* 0.01), and Omicron variant (*p* < 0.001), see Table 4.

The Alpha variant had a slightly significative higher ICU risk ratio (1.00) and ICU odds (0.03) as opposed to Omicron variant (*p* = 0.05). Similarly, but not significantly, the wild variant had a higher ICU risk ratio [0.78 (0.11–5.33)] and ICU odds (0.02) as opposed to Omicron variant (*p* 0.08).Whereas, the wild variant had a non-significant trend of higher ICU odds and ICU risk ratio from the Alpha strain (*p* 0.8). In other words, the Delta variant had the highest risk ratio and odds for access to intensive care due to severe-critical COVID-19-related symptoms compared to the other variants that dominated the respective pandemic waves (wild, Alpha, and Omicron). The Alpha strain wave caused a higher risk to the step-up unit than the Omicron wave.

### 3.3. Comparison between Vaccinated vs. Non-Vaccinated Patients during the Fourth Wave

There was a relevant, even non-statistically significative, trend (*p* = 0.06) of higher ICU access among non-vaccinated woman (Table 5) considering the population under study (N 287). 

Notably, almost two thirds (70%) of the women during the fourth wave (Omicron dominated) were fully vaccinated with at least two doses (median). During the Omicron wave, none of the vaccinated patients accessed the ICU nor needed any ventilation support or oxygen administration; only a few of them (4%) needed the use of paracetamol and IgG1k monoclonal antibodies (sotromivab, Xevudy). Notably, we found a relevant although non-statistically significant (*p* = 0.37) trend of few-mild symptomatic infection among non-vaccinated woman (22%) as opposed to vaccinated woman (15%) during the fourth wave. Pfizer was the most used vaccine (88%), whereas Moderna was administered in 11% of cases, based on vaccination hub availability, without any differences between the two groups in terms of disease severity. None of the patients during the previous pandemic waves were vaccinated.

## 4. Discussion

In this study, we focused on the prevalence and the characteristics of severe or critical cases of COVID-19 infection needing ICU admission out of 287 cases among pregnant patients. We evaluated possible differences between the following viral strains: wild, Alpha, Delta, and Omicron. We surmised from our previous results that only the severe symptoms during COVID-19 infection were responsible for worsening the obstetrical and neonatal outcomes, with higher rates of urgent and/or emergent cesarean section, preterm births, and neonatal respiratory distress syndrome [23].

New variants can take over coexisting ones, even in the case of more severe symptoms, thanks to increased transmissibility and a possible immune evasion leading to the creation of so-called VOCs. The B.1.1.7 VOC or Alpha variant spread from the end of 2020 to February 2021, when a new predominant viral strain called the Delta variant or B.1.617.2 variant emerged due to its higher transmissibility [14,24,25]. 

In November 2020, the Centers for Disease Control and Prevention (CDC) reported data on outcomes in about 326,335 women of reproductive age (15–44 years) with symptomatic COVID-19 of which 9.0% (8207) were pregnant. This report showed that pregnant women presented a significantly higher risk of ICU admission, need for mechanical ventilation and extracorporeal membrane oxygenation (ECMO), and higher death odds adjusted for age, race/ethnicity, and comorbidities [26]. At our Centre, the prevalence of severe-critical COVID-19 illness was 2.8% (8/287), reaching 4.9% (8/163) excluding the fourth wave (Omicron dominant).

Among our severe-critical cases, the Delta variant wave was associated with a higher rate of OTI and CS. Furthermore, the Alpha variant caused a higher risk to the step-up unit than the wild and Omicron variants. During the second wave (Alpha variant), as reported in current literature, the need for ventilation (OR 2.58 UKOSS study), hospital admission, and subsequent step-up admission (OR 1.62 UKOSS study; 13.5% vs. 8.9% ICNARC study), compared with the first wave (wild variant) increased among pregnant and post-partum patients with a general worsening of the disease’s severity [14,24,27,28,29,30]. An Italian population-based prospective cohort study (N 315) comparing the impact of the wild type with that of the Alpha VOC on maternal and perinatal outcomes, showed that the need for ventilatory support and/or ICU admission significantly increased during the Alpha wave [14]. A recent monocentric retrospective study showed that the prevalence of pregnant women admitted to the ICU according to the first wave (wild), second wave (Alpha and Beta dominant), and third wave (Delta dominant) was 16.5%, 21.1%, and 62.4%, respectively (*p* < 0.001) [31]. Interestingly, a population-based cohort study showed that the mean maternal mortality rates of the COVID-19 disease peaked during the initial pandemic period and further increased in the Delta pandemic period [32]. As the Delta VOC (B.1.617.2) became prevalent, the prevalence of severe or critical COVID-19 illnesses increased significantly [13] with more than 25% of women requiring hospital admission (prospective study on 1515 pregnant patients) [11] and a statistically increased overall risk for severe maternal morbidity (OR, 14.82 [95% CI, 8.77–26.4] [33]. There are few data about the Omicron wave among pregnant people with a not statistically significant (OR, 1.60 [95% CI, 0.94–2.63]) risk for any severe maternal morbidities associated [30,33], but the most important international guidelines suggest individualized delivery planning considering uterine contractility and fetal health indexes [2]. The subsequent Omicron VOC was identified in Italy at the end of November 2021. It spread widely and rapidly [25], even though most of the population (70%) completed the vaccination cycle [34], which reached two thirds of the pregnant woman at our Centre with a median of two doses during the fourth wave. At our Centre, none of the patients accessed the ICU during the fourth wave since they were mainly asymptomatic or with few-mild symptoms. We found a relevant although non-statistically significant trend of few-mild symptomatic infection among non-vaccinated woman (22%) as opposed to vaccinated woman (15%) during the fourth wave.

At the beginning of January 2021, the main gynecological, obstetrical and neonatal scientific Italian societies released a joint position paper that considered the anti-COVID-19 mRNA vaccine during pregnancy as sufficiently safe [35]. Then, the Italian Minister of Health Vaccination stated that the vaccination program should involve only pregnant women at high risk of exposure to the virus (e.g., health workers, caregivers) and/or serious complications from COVID-19 (hypertension, obesity, non-caucasic ethinicity) [36]. In May 2021, the Italian Gynecological Scientific societies updated their statement to strongly suggest the mRNA vaccine during the second and third trimester and officially encouraged the Italian Minister of Health to consider all pregnant women as fragile citizens needing priority access to the vaccination program [37]. Given the severity of Delta-related infection and the safety of mRNA vaccines without any live attenuated virus [38,39], the Italian Superior Institute of Health and then also the Minister of Health (0043293-24/09/2021-DGPRE-DGPRE-P) encouraged all pregnant woman to join the vaccination program during their second and third trimester [40,41]. The incidence of infection, as well as the risk of severe illness and hospitalization, considerably increased in the unvaccinated people compared to the vaccinated ones [19,42]. It is known that COVID-19 vaccinations in pregnant women are associated with a lower risk of maternal SARS-CoV-2 infection, neonatal intensive care admission, and stillbirth [43]. 

However, we faced an initial stiff resistance to the vaccination program that gave way to a surge in vaccinations among pregnant women only at the dawn of the fourth wave. Since the antibody titers induced by “natural” infection and/or vaccines decline with time, a booster vaccine dose five months after the primary two-dose vaccination series is needed [38,44,45]. 

Notably, all the severe-critical patients who accessed the step-up unit among our population had an AMA with a median age of 37.5 [IQR 4.5] years and a condition of mild obesity with a median BMI of 30 [IQR 4.6]. The Centers for Disease Control and Prevention (CDC) showed that pregnant women have a significantly higher risk of ICU admission, need for mechanical ventilation or ECMO, and a higher probability of death, with worse outcomes in women aged between 35 and 44 [3]. Recent studies indicate that the most relevant maternal risk factors for developing the severe-critical outcome of SARS-CoV-2 infection are as follows: AMA; high body mass index (BMI > 30); hypertensive disorders [aOR 4.3, 95% CI 1.9–9.5], especially chronic hypertension and pre-eclampsia); pulmonary comorbidities [aOR 2.7, 95% CI 1.0–7.0]; and diabetes [aOR2.2, 95% CI 1.1–4.5], mainly the pregestational one [29,30]. That 25% of our ICU patients carried a twin pregnancy might be understood as a possible adjunctive risk factor, even if this cannot still be considered a significant risk factor for severe-critical maternal outcomes in case of COVID-19 infection as previously evaluated in the literature [30]. 

Notably, our patients infected with the Delta variant had a higher rate of hyperimmune plasma infusion (75%), antibiotic therapy (75%), and remdesivir (33%); all of them were intubated. The therapeutic management of pregnant patients with COVID-19 has been considered a controversial issue because pregnant women are considered patients with an increased risk for developing severe disease, but they are traditionally excluded from the most important international trials, especially the ones regarding the use of newly discovered drugs [2,46]. The American College of Obstetricians and Gynecologists (ACOG) developed an algorithm to evaluate and manage pregnant outpatients with COVID-19, however similar to nonpregnant women [2]. Systemic corticosteroid therapy (especially dexamethasone) for ten days or up to discharge improves clinical outcomes and reduces mortality in pregnant hospitalized patients who need supplemental oxygen and/or mechanical ventilation [2,46]. Antibiotic therapy is not currently considered a “standard” therapy, even though macrolides can be usefully administered in the “early phases” of the disease. Hyperimmune plasma is a second-line treatment for severe-critical forms of COVID-19 (major hypoxemia and need for mechanical ventilation), even if little is known of its use during pregnancy; two units of plasma should be “early” used (median of two days from hospitalization) due to its therapeutical time-linked anti-viral effect [47]. Small studies reported the use of the antiviral drug remdesivir during pregnancy without adverse effects [2]. The most important international guidelines suggest not to stop this drug in these populations without a strict and serious reason [2]. Intravenous remdesivir is approved worldwide for the treatment of COVID-19 in adult and pediatric patients of mild to moderate COVID-19 in high-risk, non-hospitalized patients [2]. However, remdesivir should only be administered to pregnant women who are showing a lack of improvement or deterioration from COVID-19 symptoms [46]. In the case of mild symptoms without the need for COVID-19 related hospitalization, the use of IgG1k monoclonal antibodies (sotromivab, Xevudy) is strongly suggested for pregnant and breastfeeding woman, especially with risk factors (being unvaccinated, Black or Asian ethnicity, BMI > 25 kg/m^2^, diabetes or hypertension, AMA, socioeconomic deprivation, working in healthcare or other public-facing occupations [46,48]), as in our population during the last pandemic wave after informed consent.

## 5. Conclusions

This monocentric and cross-sectional study estimated the prevalence (2.8%) of intensive care unit access among pregnant women (N 287) with a significantly increased risk ratio during the second wave (3%; dominated by Alpha) and especially the third wave (25%, dominated by Delta) of the COVID-19 pandemic. 

The maternal outcome worsened in the case of the Alpha variant, and especially the Delta variant for severe-critical COVID-19-related symptoms and ICU access. The absence of severe-critical cases during the fourth wave (Omicron dominant) might be explained by the high diffusion of mRNA-based vaccination. Thus, vaccination during the second and third trimester of pregnancy against COVID-19 is strongly recommended.

Our study had the following limitations: small sample size for severe-critical cases (monocentric study); the determination of the variant on hospitalized patients (by sample and not systematic for each of them) creating a sampling bias; none of the patients during the wild, Alpha, and Delta waves were vaccinated. 

## Figures and Tables

**Figure 1 vaccines-10-02061-f001:**
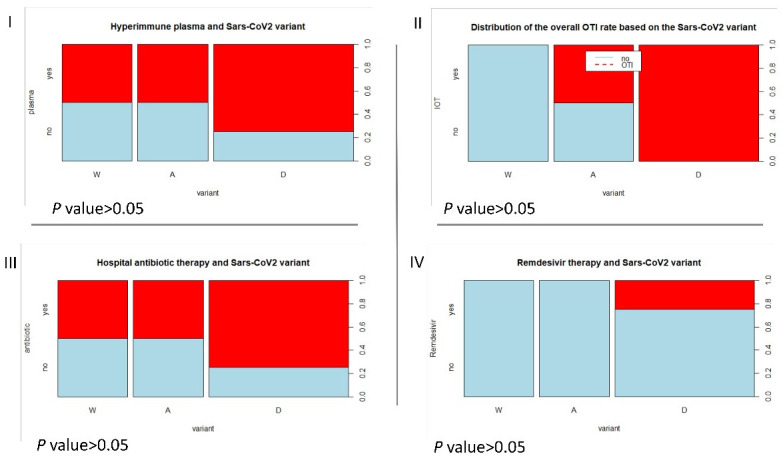
(**I**) The different therapy performed (in red) during the step-up stay of severe-critical COVID-19 pregnant women based on the variant (W = wild, A = alpha, D = delta). Half of W and A underwent hyperimmune plasma and antibiotic therapy, whereas in D the percentage rose to 75%. (**II**) None of the W patients underwent OTI, whereas half of the A and all the D patients needed it. (**III**) the different therapy performed (in red) during the step-up stay of COVID-19 affected pregnant women based on the variant (W = wild, A = alpha, D = delta). (**IV**) None of the W and A patients underwent remdesivir, whereas 33% of D needed it. The graphs represent a descriptive comparison of the variables’ proportions.

**Figure 2 vaccines-10-02061-f002:**
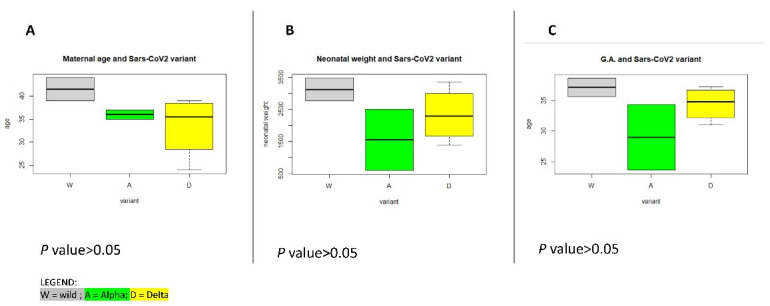
From left to right (**A**–**C**), the different characteristics of severe-critical COVID-19 affected pregnant women based on the variant (W = wild in grey, A = alpha in green, D = delta in yellow). From left to right: the maternal age, the gestational age (g.a.) and the neonatal weight. The median maternal age and the median gestational age (g.a.) were higher in the wild group, respectively 41.5 [2.5] yrs and 37.1 [1.5] wks, ds. Notably, the A group was characterized by the lowest median g.a., in detail 28.9 [5.35] wks, ds, and the lowest neonatal weight, 1145 [955] g. See Table 2.

**Table 1 vaccines-10-02061-t001:** General characteristics of the population (N 287).

Age (years)	<30	31.15%
	30–40	62.30%
	>40	5.74%
	Missing	0.82%
BMI	<25 normal and under weight	16.39%
	25–30 overweight	56.56%
	>30 obese	20.49%
	missing	6.56%
Ethnicity	Caucasian	95.08%
	African	4.10%
	Asian	0.82%
Comorbidities	hypertension/pre-eclampsia	1.64%
	diabetes/gestational diabetes	9.84%
	Others	23.77%
	Healthy	64.75%
	Missing	0.82%
Parity	Nulliparous	40.98%
	Multiparous	59.02%
G.A. at Covid diagnosis	1° trimester	0.00%
	2° trimester	1.64%
	3° trimester	98.36%
COVID-19 SYMPTOMS	1 none	60.66%
	2A few	19.67%
	2B mild	14.75%
	3 severe-critical	4.10%
	Missing	0.82%

**Table 2 vaccines-10-02061-t002:** Main characteristics of the severe-critical cases (N 8).

AGE	37.5 [4.5] *	
BMI	30 [4.6] *	
COMORBITIDY	100% none	
ETHNICITY	100% Caucasic	
SARS-CoV2 vaccination	0%	
SARS-CoV2 variant	Wild Alpha Delta Omicron	25% 25% 50% 0%
Epidemic wave	I II III IV	25% 25% 50% 0%
COVID-19 detection	100% nasopharyngeal swab, RT-PCR analysis	
Step-up days stay	13.50 [7.75] * days	
Overall OTI	57%	
Overall BiPAP	75%	
Hyperimmune plasma adm.	62%	
Remdesivir adm.	12%	
Antibiotics adm.	62%	
Heparin adm.	88%	
Mode of delivery	100% CS	
Primigravida	50%	
Twin pregnancy	25%	
G.A.	34.9 [3.72] *	
Neonatal weight	2570 [1115] * g	
Neonatal weight percentile	56.5 [34] *	
Neonatal step-down	27%	
Neonatal step-up	18%	

Legend: Comorbidities were thyroid diseases, autoimmune diseases, hereditary thrombosis, diabetes, and other chronic diseases. * Next to the median value of numerical variables, is reported the interquartile (IQR) as […]. IQR (x) = quantile (x, 3/4) − quantile (x, 1/4).

**Table 3 vaccines-10-02061-t003:** Different maternal and obstetrical outcomes based on the SARS-CoV2 variant.

	Wild	Alpha	Delta
AGE	41.5 [2.5] * yrs	36 [1] * yrs	35.5 [7.5] * yrs
Overall OTI	0%	50%	100%
Hyperimm. plasma	50%	50%	75%
Remdesivir	0%	0%	33%
Antibiotic therapy	50%	50%	75%
Neonatal weight	3125 [355] * g	1145 [955] * g	2290 [1017.5] * g
G.A.	37.1 [1.5] * wks, ds	28.9 [5.35] * wks, ds	34.7 [3.72] * wks, ds

Legend: Next to the median value of numerical variables is reported the interquartile (IQR) as […] *. IQR (x) = quantile (x, 3/4) – quantile (x, 1/4). None of the differences among the viral strains were statistically significant (*p* > 0.05).

**Table 4 vaccines-10-02061-t004:** Risk ratio estimation and confidence intervals for admission to the step-up unit (ICU) for severe-critical COVID-19 health issues based on the different viral strain that dominated each pandemic wave.

Variant	No ICU Admission	ICU Admission	Odds for ICU Admission	Risk Ratio (95% CI)
W	78	2	0.02	0.78 (0.11–5.33)
A	61	2	0.03	1.00
D	16	4	0.25	6.30
O	124	0	0.00	0.00
**Groups compared in rx2 tables**				***p*-value of the risk ratio** **comparison**
W vs. A				0.8
**W vs. D**				**0.01**
W vs. O				0.08
**A vs. D**				**0.01**
**A vs. O**				**0.05**
**D vs. O**				**<0.001**

Legend: In bold are the statistically significative results. We calculated the risk ratio by unconditional maximum likelihood estimation (Wald), and small sample adjustment (small) out of 287 pregnant patients depending on the SARS-CoV2 variant (W = wild, A = alpha, D = delta). Each row of the rx2 table was compared to the ICU access with reference level and test of independence two-sided *p* values were calculated using Fisher’s Exact, Monte Carlo simulation, and the chi-square test.

**Table 5 vaccines-10-02061-t005:** Risk ratio estimation and confidence intervals for admission to the step-up unit (ICU) for severe-critical COVID-19 health issues based on vaccination status.

Vaccine	No ICU Admission	ICU Admission	Odds for ICU Admission	Risk Ratio (95% CI)
Yes	87	0	0.00	1.00
No	192	8	0.04	0.00
**Groups compared in rx2 tables**				***p*-value of the risk ratio ** **comparison **
Vaccinated vs. Non-vaccinated				0.06

Legend: We calculated the risk ratio by unconditional maximum likelihood estimation (Wald), and small sample adjustment (small) out of 287 pregnant patients depending on their COVID-19 vaccination status (at least two doses to be considered fully vaccinated). Each row of the rx2 table was compared to the ICU access with reference level and test of independence two-sided *p* values were calculated using Fisher’s Exact, Monte Carlo simulation, and the chi-square test.

## Data Availability

Not applicable.

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
