# Peer review of "Adverse Maternal Outcomes in Pregnant Women Affected by Severe-Critical COVID-19 Illness: Correlation with Vaccination Status in the Time of Different Viral Strains’ Dominancy"

_vaccines, 2022, doi:10.3390/vaccines10122061_

Round 1
Reviewer 1 Report
Introduction: 1st paragraph, line no: 40. Grammatical error, quickly spread*
Introduction needs to describe briefly. At this time, number of literature are bussing with the evidence. Authors failed to cite some the key references. There is no literature gap has found. First paragraph can be shortened. Keep abbreviation consistent i.e COVID-19, some places authors have written covid-19. either keep it in big or small letter.
Method: Study was supposed to be intended to correlate outcome with vaccinated individuals as well. As i couldn't see the statement in inclusion criteria. For Delta and Omicron authors didn't confirm the sample. Therefore, it could be consider as sampling bias which needs to be mentioned clearly in study Limitation section. Study aim needs to move to introduction section. Instead, only mention primary and secondary end points in method.
Results: Kindly spelled out abbreviation in table section. Sample size is very low to form a statement regarding clinical outcome difference. kindly mention which statistical test performed in figure/chart.
Kindly redraft the manuscript and clearly state the study intention.
Author Response
Comments and Suggestions for Authors
Introduction: 1st paragraph, line no: 40. Grammatical error, quickly spread*
Thank you, we made the correction.
Introduction needs to describe briefly. At this time, number of literature are bussing with the evidence. Authors failed to cite some the key references. There is no literature gap has found. First paragraph can be shortened.
Thank you, we made the correction shortening the paragraph and adding more references.
Keep abbreviation consistent i.e COVID-19, some places authors have written covid-19. either keep it in big or small letter.
Thank you, we made the correction.
Method: Study was supposed to be intended to correlate outcome with vaccinated individuals as well. As i couldn't see the statement in inclusion criteria. For Delta and Omicron authors didn't confirm the sample. Therefore, it could be consider as sampling bias which needs to be mentioned clearly in study Limitation section. Study aim needs to move to introduction section. Instead, only mention primary and secondary end points in method.
Thank you, we made the corrections and we added the missing information.
Results: Kindly spelled out abbreviation in table section. Sample size is very low to form a statement regarding clinical outcome difference. kindly mention which statistical test performed in figure/chart.
Thank you, we made the corrections and we added the missing information.
Kindly redraft the manuscript and clearly state the study intention.
Thank you, we redrafted the manuscript with the corrections.
Reviewer 2 Report
I was invited to revise the paper entitled "Adverse maternal outcomes in pregnant women affected by severe-critical Covid-19 illness: correlations to different viral strains of SARS-CoV-2 and to vaccinated/not vaccinated pregnant women". It was a cross-sectional study performed in a Southern Italian Hospital. The topic is interesting and the study was conducted rigorously. I have some major observations:
- Statistical analysis section should be improved. Which was the model choosen to estimate the risk-ratio?
- Authors shoul present firstly a table with baseline characteristics of all included patients;
- All table titles were totally inappropriate. Use a descrition title for each table and shift the description under the table;
- How did Authors considered the vaccination status or previous infections? how did they managed it in the risk models?
- Table 3 is totally unreadable and it unclear the type of comparison performed;
- Figure 1 and 2 have low quality. Pleas improve them and add a p-value for the comparison among groups;
- Authors should also compare outcomed between prevaccination era and post-vaccination era, in order to evaluate the change in pregnancy outcomes by vaccination status and VOC.;
- About vaccination, Authors should evaluate differences among different type of vaccine.
Author Response
I was invited to revise the paper entitled "Adverse maternal outcomes in pregnant women affected by severe-critical Covid-19 illness: correlations to different viral strains of SARS-CoV-2 and to vaccinated/not vaccinated pregnant women". It was a cross-sectional study performed in a Southern Italian Hospital. The topic is interesting and the study was conducted rigorously. I have some major observations:
- Statistical analysis section should be improved. Which was the model choosen to estimate the risk-ratio?
We calculated the risk ratio by unconditional maximum likelihood estimation (Wald), and small sample adjustment (small). Confidence intervals are obtained by using normal approximation (Wald), and normal approximation with small sample adjustment (small), and bootstrap method (boot). Similarly, each row of the rx2 table is compared to the exposure (i.e. ICU yes) reference level and test of independence two-sided p values are calculated using Fisher's Exact, Monte Carlo simulation, and the chi-square test.
- Authors should present firstly a table with baseline characteristics of all included patients;
Thank you, we added the missing information.
- All table titles were totally inappropriate. Use a descrition title for each table and shift the description under the table;
Thank you, we made the corrections and we added the missing information.
- How did Authors considered the vaccination status or previous infections? how did they managed it in the risk models?
We did not include the vaccination status in the risk model for ICU access since the vaccination started from the omicron wave that did not experience any ICU access. The previous infection cases were registered during the omicron wave for woman with a previous infection. None of the wild, alpha or delta wave reported a previous infection.
- Table 3 is totally unreadable and it unclear the type of comparison performed;
Thank you, we made the corrections and we added the missing information.
- Figure 1 and 2 have low quality.
The high-resolution images will be uploaded apart from the manuscript file, as indicated in the author’s guidelines.
Pleas improve them and add a p-value for the comparison among groups;
Thank you, we made the corrections and we added the missing information.
- Authors should also compare outcomed between prevaccination era and post-vaccination era, in order to evaluate the change in pregnancy outcomes by vaccination status and VOC.;
Line 391 ….none of the patients affected by Covid-19 during the w, α, and δ waves were vaccinated, thus the absence of severe-critical cases during the 4th wave (Omicron dominant) was caused by possible lower morbidity or by the diffusion of mRNA-based vaccination.
Line 283-288 … Pfizer was the most used vaccine (88%), whereas Moderna was administered in 11 % of cases without any differences about the two groups for disease severity. Notably, we found a non-significant (p=0.37) trend of symptomatic infection among non-vaccinated woman (22%) vs vaccinated woman (15%) comparing vaccinated vs non vaccinated woman during the Omicron wave. None of the patients during the previous pandemic waves were vaccinated.
- About vaccination, Authors should evaluate differences among different type of vaccine.
Line 283-284 … Pfizer was the most used vaccine (88%), whereas Moderna was administered in 11 % of cases without any differences about the two groups for disease severity.
Reviewer 3 Report
This is a cross-secional study among 287 pregnant patients who required admission into the Intensive Care Unit due to severe Covid-19. The correlations between the SARS-CoV-2 variants and the adverse maternal outcomes are presented. The delta variant associates the highest risk and odds for access to ICU, compared with other variants. The omicron wave associates scarce symptoms. The main pitfall is the limited number of pregnant patients included in the study. The main interest of the study is the comparison of maternal outcomes of pregnant patients with Covid-19 according to the different SARS-CoV-2 variants.
Author Response
Thank you, we underlined the main pitfall in the limitations section.
Round 2
Reviewer 1 Report
Dear Authors,
I have gone through the revised version of submitted manuscript. Manuscript drafting is scientifically not in line with. There are multiple limitation found in manuscript.
Author Response
Reject report:
"I have gone through the revised version of submitted manuscript. Manuscript
drafting is scientifically not in line with. There are multiple limitation
found in manuscript:
First, Article title isn't appropriate as authors mentioned correlation to
different viral strain, though, authors didn't confirm the strain. It was an
assumption based on variants dominancy. Instead, redraft as a Correlation in
the time of different viral strains dominancy.
Thank you, we changed the title in “ Adverse maternal outcomes in pregnant women affected by severe-critical Covid-19 illness: correlation with vaccination status in the time of different viral strains’ dominancy.”
2nd, Introduction section is still poor, authors failed to cite some of the
key references. Literature are buzzing with the evidence. So, no literature
gap has found.
Thank you, we studied many other papers that we cited in the text.
3rd, Study design isn't enough to conclude.
Thank you, we are planning another study with a wider sample size. We admitted the limitations of this monocentric study.
4th, From statistical perspective, I couldn't find any insightful evidence.
at least i would expect, i.e Comparison between non vaccinated vs vaccinated,
Multivariate regression, significance, association.
Thank you, we added comparison between vaccinated and non-vaccinated with p-value.
5th, discussion is not in flow. failed to cite some of the key references.
Thank you, we studied many other papers that we cited in the text.
Limitation haven't clearly stated."
Thank you, we clarified the limitations.
Reviewer 2 Report
I was invited to review the revised version of the paper entitled "Adverse maternal outcomes in pregnant women affected by severe-critical Covid-19 illness: correlations to different viral strains of SARS-CoV-2 and to vaccinated/not vaccinated pregnant women".
Authors addressed all comments. I have only some observations:
- Risk models should be described in the Statistical Analysis section;
- Authors stated: " thus the absence of severe-critical cases during the 4th wave (Omicron dominant) was caused by possible lower morbidity or by the diffusion of mRNA-based vaccination.". References? This sentence need to be confirmed. Probably the vaccination status in the main reason of this point.
- Why did Authors stated that "We did not include the vaccination status in the risk model for ICU access since the vaccination started from the omicron wave"? Vaccination campagne in Italy started in January 2021. Omicron wave was referred to November 2021, so this sentence was totally wrong. Vaccination status need to be evaluated in the risk model. If Authors did not have information about vaccination status, the model is totally useless.
Author Response
I was invited to review the revised version of the paper entitled "Adverse maternal outcomes in pregnant women affected by severe-critical Covid-19 illness: correlations to different viral strains of SARS-CoV-2 and to vaccinated/not vaccinated pregnant women".
Authors addressed all comments. I have only some observations:
- Risk models should be described in the Statistical Analysis section;
Thank you, we added the section of risk models.
“We calculated the risk ratio by unconditional maximum likelihood estimation (Wald), and small sample adjustment (small) out of 287 pregnant patients related depending on the SARS-CoV2 variant (W=wild, A=alpha, D=delta). Confidence intervals are obtained by using normal approximation (Wald), and normal approximation with small sample adjustment (small), and bootstrap method (boot). Similarly, each row of the rx2 table is compared to the exposure (i.e. ICU yes) reference level and test of independence two-sided p values are calculated using Fisher's Exact, Monte Carlo simulation, and the chi-square test.”
- Authors stated: " thus the absence of severe-critical cases during the 4th wave (Omicron dominant) was caused by possible lower morbidity or by the diffusion of mRNA-based vaccination.". References? This sentence need to be confirmed. Probably the vaccination status in the main reason of this point.
Thank you, it was a speculation therefore we clarified the hypothetical sentence.
- Why did Authors stated that "We did not include the vaccination status in the risk model for ICU access since the vaccination started from the omicron wave"? Vaccination campagne in Italy started in January 2021. Omicron wave was referred to November 2021, so this sentence was totally wrong. Vaccination status need to be evaluated in the risk model. If Authors did not have information about vaccination status, the model is totally useless.
Thank you, we clarified this point by adding more information about the Italian vaccination campaign among pregnant women. The vaccination campaign did not include all the pregnant woman at that time, but only who belonged to special work categories and there were many antivaxxers among patients.
“At the beginning of January 2021, the main gynecological, obstetrical and neonatal scientific Italian societies released a joint position paper that considered as sufficiently safe the anti-Covid19 mRNA vaccine during pregnancy [29]. In May 2021, they updated their statement that strongly suggested the mRNA vaccine during the second and third trimester officially encouraging the endorsement by the Italian Minister of Health to consider the pregnant women as fragile citizens needing a priority access to the vaccination program [30]. Since the severity of Delta related infection and the safety of mRNA vaccines without any live attenuated virus [21, 31], the Italian Superior Institute of Health, and then also the Mister of Health (0043293-24/09/2021-DGPRE-DGPRE-P), encouraged all the pregnant woman, not only the ones at professional risk (working in healthcare or other public-facing occupations), during the second and third trimester to join the vaccination program [32, 33]. However, we faced a stiff resistance to the vaccination program.”
Since the severity of Delta related infection and the safety of mRNA vaccines without any live attenuated virus [21, 29], the Italian Superior Institute of Health encouraged all the pregnant woman, not only the ones at professional risk (working in healthcare or other public-facing occupations), during the second and third trimester to join the vaccination program [30]. The subsequent Omicron VOC was identified in Italy at the end of November 2021 and then it widespread rapidly [19], even though most of the population completed the vaccination cycle [31])”
Moreover, we added this paragraph in the results section and table 5:
3.3 Comparison between vaccinated vs non-vaccinated patients during the fourth wave.
“….. Notably, we found a relevant although non-statistically significant (p=0.37) trend of few-mild symptomatic infection among non-vaccinated woman (22%) vs vaccinated woman (15%) during the 4th wave. …… There was also a non-significative trend of higher ICU access among the non-vaccinated woman (table 5) considering all the population under study (N 287).
Round 3
Reviewer 2 Report
The paper can be now accepted